# Biomimetic Design and Validation of an Adaptive Cable-Driven Elbow Exoskeleton Inspired by the Shrimp Shell

**DOI:** 10.3390/biomimetics10050271

**Published:** 2025-04-28

**Authors:** Mengqian Tian, Yishan Liu, Zhiquan Chen, Xingsong Wang, Qi Zhang, Bin Liu

**Affiliations:** 1School of Mechanical Engineering, Southeast University, Nanjing 211189, China; ysliu@seu.edu.cn (Y.L.); zhiquanchen@seu.edu.cn (Z.C.); xswang@seu.edu.cn (X.W.); 2School of Automation, Nanjing University of Information Science and Technology, Nanjing 210044, China; qizhang_mech@nuist.edu.cn; 3Suzhou Institute of Biomedical Engineering and Technology, Chinese Academy of Sciences, Suzhou 215163, China

**Keywords:** elbow exoskeleton, cable driven, shrimp shell, biomimetic design, PID control, rehabilitation

## Abstract

The application of exoskeleton robots has demonstrated promising effectiveness in promoting the recovery of motor skills in patients with upper limb dysfunction. However, the joint misalignment caused by rigid exoskeletons usually leads to an uncomfortable experience for users. In this work, an adaptive cable-driven elbow exoskeleton inspired by the structural characteristics of the shrimp shell was developed to facilitate the rehabilitation of the elbow joint and to provide more compliant human-exoskeleton interactions. The exoskeleton was specifically designed for elbow flexion and extension, with a total weight of approximately 0.6 kg. Based on the mechanical design and cable configuration of the exoskeleton, the kinematics and dynamics of driving cables were analyzed. Subsequently, a PID-based control strategy was designed with cable kinematics. To evaluate the practical performance of the proposed exoskeleton in elbow assistance, a prototype was established and experimented with six subjects. According to the experimental results, the measured elbow joint angle trajectory is generally consistent with the desired trajectory, with a mean position tracking accuracy of approximately 0.997, which supports motion stability in rehabilitation scenarios. Meanwhile, the collected sEMG values from biceps brachii and brachioradialis under the exoskeleton condition show a significant reduction in average muscle activation by 37.7% and 28.8%, respectively, compared to the condition without exoskeleton.

## 1. Introduction

Upper limb dysfunction may arise from various medical conditions, such as neurodegenerative diseases [1], cerebrovascular injuries [2], and spinal cord injuries [3], potentially exerting a substantial negative influence on the daily life of affected individuals. Rehabilitation training is widely used to improve the recovery of continuous functions, with passive limb movement serving as a primary therapeutic approach. Repetitive movement of muscles and joints has been demonstrated to facilitate the reorganization and reconstruction of residual neural pathways. However, due to the limited human resources available for rehabilitation therapy and the increased risk of secondary injuries caused by subjective judgment during treatment, the application of robotic exoskeletons for rehabilitation is highly necessary.

Currently, a variety of exoskeletons have been developed for assisting movements of different joints, employing various actuation methods, structural designs, and control strategies. The effectiveness of exoskeleton-assisted training in improving upper limb function has been validated [4]. Among these, rigid exoskeletons have emerged as the predominant approach. For instance, Otten et al. designed a hydraulically driven self-aligning upper limb exoskeleton for stroke rehabilitation [5]. Kim et al. investigated a seven degrees-of-freedom upper limb exoskeleton, in which each joint is driven by a brushless motor coupled with a harmonic drive [6]. Pan et al. presented a shoulder-elbow exoskeleton, achieving four degrees of freedom through serial actuators [7]. However, the excessive driving forces generated by rigid exoskeletons can compromise safety and comfort during human-robot interaction. To address this, incorporating more compliant actuation strategies or structural designs is essential to improve the overall user experience.

To enhance the adaptability and conformability of exoskeletons, researchers developed soft exoskeletons. These systems employ flexible materials, such as fabric-based wearable suits and soft actuators. The soft materials are more suitable for contact with the human body, and the soft actuators generally exhibit lighter weights and lower impact forces, significantly improving comfort and safety during human-robot interactions. Pneumatic artificial muscles are among the most commonly utilized soft actuators, mimicking the contraction and relaxation of biological muscles via air pressure. McKibben pneumatic muscles were applied in many soft exoskeletons, where inflation causes the airbag to expand, producing linear contraction. The radial pulling force generated assists the required joint rotation [8,9]. However, these actuators can only provide pulling force and require antagonistic muscle pairs or additional structures to facilitate return movement. In addition, pneumatic actuators made from soft textiles are also widely used. Pneumatic actuators with special shapes deform upon inflation, thereby supporting joint movement. Notable examples include a soft pneumatic elbow exoskeleton investigated by John et al. [10], soft robotic elbow sleeves designed by Koh et al. [11], a high-force soft actuator presented by Benjamin et al. [12], a new soft elbow exosuit researched by Thalman et al. [13], and a pneumatic actuator integrated with sensors through machine weaving designed by Luo et al. [14]. Although soft exoskeletons typically offer advantages in human-robot interactions, comfort, and safety, their actuation force remains insufficient for effective rehabilitation, particularly for patients with increased joint stiffness.

Many exoskeletons employ Bowden cable-driven systems, which are characterized by their lightweight structure, high safety, and excellent flexibility. These features make them particularly well-suited for applications in the field of rehabilitation. Some research has integrated Bowden cables with rigid structures to leverage the advantages of both flexible transmission and structural support, such as Armule [15], ALEx [16], and an upper-limb exoskeleton designs by Wang et al. [17]. These exoskeletons typically consist of two or more rigid links affixed to the upper arm and forearm, with a revolute joint mimicking human elbow articulation. The rotation of this joint drives the motion of the forearm along a fixed axis, enabling elbow flexion and extension. However, this kind of configuration imposes extremely stringent requirements on the installation of the exoskeleton, necessitating the precise match of the revolute joint axis of the exoskeleton with that of the human elbow, which poses significant challenges. Furthermore, the relative displacement between the exoskeleton and the user during movement induced by external forces exacerbates misalignment problems. To combine the advantages of soft exoskeletons with sufficient actuation force, many researchers have integrated Bowden cables into soft wearable fabric straps. For elbow exoskeletons, anchor points are typically placed on the upper arm and forearm, allowing the forearm to rotate around the elbow joint via cable contraction [18,19,20,21,22,23,24]. These designs offer lightweight and easily wearable solutions. However, the soft suit or straps are prone to displacement under external forces, which may compromise rehabilitation efficacy. Additionally, the conventional parallel cable layout, in which anchor points are distributed directly along the arm, presents limitations. During elbow flexion, cable tension can increase the distance between the cable and the upper surface of the arm, reducing human-robot conformability and increasing potential safety risks, such as cable entanglement with external objects.

Given the current research status, an adaptive cable-driven elbow exoskeleton inspired by the shrimp shell was designed for elbow rehabilitation of patients with impaired mobility. The multi-modular shrimp shell structure ensures that the exoskeleton can adaptively adjust with joint movement, thereby improving compliance during human-exoskeleton coupling interactions. The rest of this paper is organized as follows: The mechatronic system of the exoskeleton is introduced in Section 2, the kinematics and dynamics of actuation cables are analyzed in Section 3, the applied motion control strategy is described in Section 4, experiments are conducted and results are discussed in Section 5, and all the presented research contents and results are summarized in Section 6.

## 2. Exoskeleton Description

### 2.1. Mechatronic System

To enhance compliance and comfort in human-robot interaction while ensuring motion accuracy of the elbow joint to prevent secondary injury to patients, a shrimp-inspired elbow exoskeleton system has been designed. The elbow rehabilitation training of a patient with the proposed exoskeleton system is schematically illustrated in Figure 1. Given that the target application is rehabilitation training for patients with upper limb dysfunctions, emphasis is placed on developing a lightweight and reliable exoskeleton rather than prioritizing portability. Therefore, the elbow exoskeleton and cable actuator are installed on a fixed frame. For elbow rehabilitation training, patients should sit on a chair or bench while wearing the exoskeleton on their upper arm and forearm to assist with elbow flexion and extension movements. Here, the cable actuator is designed to utilize a DC brushless motor, which transmits the driving force to the exoskeleton joint via Bowden cables. The Bowden cable consists of a high-strength steel wire encapsulated within a flexible outer sheath, which provides structural support and protection while permitting the inner wire to slide internally, thereby transmitting both force and mechanical displacement. The flexible Bowden cable demonstrates significant bendability and adaptability to complex routing configurations, while its lightweight design substantially enhances the wearability and user experience.

The hardware architecture of the cable-driven actuator comprises three primary components, including a microcontroller unit, a brushless DC motor, and sensors. The brushless DC motor, equipped with an integrated gear reducer, is directly connected to the active reel. The reels are configured with helical grooves to wind the cable. The passive reel moves through a pair of gears, allowing a single motor to drive two cables. Due to the initial position of the elbow exoskeleton, where the arm naturally hangs, the driving cables continuously exert a force on the forearm throughout the movement to overcome gravity. As a result, the driving cables remain taut at all times, preventing them from slipping out of the reel due to slackening.

The elbow exoskeleton, designed with an adaptive biomimetic structure inspired by the shrimp shell, weighs approximately 0.6 kg and features two drive cables arranged in an inclined cross pattern. The upper arm cuff (UAC) of the exoskeleton is fixed to the frame, while the forearm is secured to the forearm cuff (FAC) using an airbag integrated into the FAC. The airbag adapts to the varying limb dimensions of different users, while the inflation pressure provides a secure attachment between the exoskeleton and the limb. Before starting rehabilitation training, both the UAC and FAC should be properly secured to the upper arm and forearm of the patient. The elbow exoskeleton acquires angular data through inertial measurement units (IMUs) mounted on the UAC and FAC. The difference between the angular measurements of the two IMUs represents the angle of the elbow joint. The tensile forces on the driving cables are measured using force sensors.

Conventional cable routing configurations for elbow joint actuation typically employ vertically aligned anchor points positioned along the central axis of the arm. However, during the elbow flexion movement, the driving cable remains taut, forming a distinct triangular relationship with the forearm, which increases the risk of the cable becoming entangled with foreign objects and reduces the human-robot conformity. To address these limitations, the present study implements an oblique cable routing strategy with two cables, S1 and S2. The comparison of single vertical and double oblique cable layouts is illustrated in Figure 2. By offsetting both the anchor points and cable exit points from the central axis, positioning them closer to the dorsal surface of the limb rather than at the fixation apex of the device, the oblique cable configuration demonstrates superior conformity to human anatomy throughout the range of motion. In contrast to the single-cable layout, the dual diagonal cables operate in coordination to enhance the stability of the motion system. The symmetrical configuration allows the lateral tensile forces to mutually cancel out, thereby effectively preventing lateral deviation or torsional motion of the exoskeleton during operation. This kind of cable arrangement also contributes to improving control accuracy.

To distribute the mechanical forces at the anchor points, a double figure eight loop is employed to form multiple anchor points, which are evenly secured to the FAC. To facilitate stable return movement of the exoskeleton, an elastic cord is mounted on the posterior aspect of the exoskeleton. This cord utilizes elastic restoring force to provide antagonistic action during forearm extension, ensuring a smooth return to the initial position following flexion. Furthermore, the elastic properties of the cord effectively absorb impact and vibration during motion, thereby enhancing both stability and comfort during operation.

### 2.2. Biomimetic Design

To provide support force between the cable anchor points and redirection points and prevent relative displacement during dynamic motion, a multi-modular adaptive structure inspired by the shrimp shell was designed. Shrimp, as crustacean arthropods, possess a chitin-based exoskeleton with segmented body architecture. Their abdomen is typically composed of six distinct segments, which are interconnected through articular membranes, providing specific degrees of freedom for the movement of individual joints [25,26]. By precisely controlling the angular variations between adjacent segments, the shrimp can execute bending and extension movements with varying amplitudes. The motion flexibility and adaptive capability of the elbow exoskeleton can be achieved by drawing inspiration from the multi-modular structure of the shrimp exoskeleton.

The shrimp-inspired adaptive elbow exoskeleton is illustrated in Figure 3. The exoskeleton design mainly comprises six interconnected modules, analogous to the segmented abdomen of a shrimp. To replicate the specific angular mobility observed in the shrimp shell, each module is interconnected in a manner that allows controlled rotational movement. Adjacent modules are secured by screws positioned at their rotational centers, while the inclined edges of each module serve to constrain the range of motion at each joint. Adjacent modules maintain substantial contact surface area, ensuring axial abutment and effectively preventing relative displacement during motion. During elbow flexion, this configuration allows the exoskeleton to autonomously adjust its axis of rotation, thereby improving motion flexibility and human-exoskeleton coordination. The joint interface between modules A and B features complementary protrusions and recesses that interlock to form a revolute pair, providing axial support and precise positioning. To prevent misalignment of protrusions and recesses, bilateral limiters are incorporated on both sides of the recessed modules.

To better accommodate the motion patterns of the elbow joint, the rotational angles between adjacent modules were rationally designed. The dented module IV, aligned with the elbow joint, features larger rotational angles of 65° on both sides. The rotational angles gradually decrease in modules further from the elbow joint. The overall structure provides a rotational range of 0–140°, fully accommodating the flexion range of the elbow. To enhance wearer comfort and achieve lightweight requirements, the structure was fabricated using resin material through 3D printing technology.

## 3. Cable-Driven Analysis

Since the driving force of the motor is not directly exerted on the forearm of the patient but is transmitted indirectly via cable retraction to generate forearm motion, it is crucial to establish an accurate model between the actuation space and the joint space. This model is realized through the kinematic modeling of the exoskeleton, enabling the transformation of cable length variations into corresponding changes in the angular displacement of the elbow joint.

### 3.1. Cable Kinematics

In this study, a kinematic model of human elbow flexion-extension motion was developed to mathematically characterize the relationship between joint angular displacement and cable length variation. In the exoskeleton system, as illustrated in Figure 4a, the cable length is defined as the linear distance between the redirection points (B1 and B2) on the UAC and the cable anchor points (A1 and A2) on the FAC.

Points O1 and O2 are the center points of FAC and UAC, respectively. $r_{1}$ represents the position vector of A1 relative to O1, while $r_{2}$ represents the position vector of B1 relative to O2. $p_{1}$ and $p_{2}$ represent the position vectors of A1 and A2 relative to the rotational center of the elbow joint (E), while $l_{1}$ and $l_{2}$ represent the position vectors of B1 and B2 relative to E. Subsequently, the projection onto the YOZ plane was modeled, as depicted in Figure 4b. Due to the symmetrical distribution of the two cables with identical displacements, only S1 is taken as an example. In the schematic representation, $r_{1YOZ}$ and $r_{2YOZ}$ represent the projections of vectors $r_{1}$ and $r_{2}$ onto the YOZ plane. In addition, $b_{1}$ and $b_{2}$ correspond to the distances from O1 and O2 to E, respectively. $p_{1YOZ}$ and $l_{1YOZ}$ represent the projections of vectors $p_{1}$ and $l_{1}$ onto the YOZ plane, respectively. By leveraging geometric relationships, the following equations were derived:(1)$$p_{1YOZ}=\sqrt{r_{1YOZ}^{2}+b_{1}^{2}}$$(2)$$l_{1YOZ}=\sqrt{r_{2YOZ}^{2}+b_{2}^{2}}$$(3)$$\theta_{1}=\arctan\left(\frac{r_{1YOZ}}{b_{1}}\right)$$(4)$$\theta_{2}=\arctan\left(\frac{r_{2YOZ}}{b_{2}}\right)$$(5)$$\beta =180^{\circ }-\left(\alpha +\theta_{1}+\theta_{2}\right)$$

By the law of cosines, it can be obtained that:(6)$$\begin{array}{cc} X(\alpha ) & =\sqrt{p_{1YOZ}^{2}+l_{1YOZ}^{2}-2p_{1YOZ}l_{1YOZ}\cos\left(\beta \right)}\\ \; & =\sqrt{p_{1YOZ}^{2}+l_{1YOZ}^{2}-2p_{1YOZ}l_{1YOZ}\cos\left(180^{\circ }-\left(\alpha +\theta_{1}+\theta_{2}\right)\right)} \end{array}$$(7)$$L\left(\alpha \right)=\sqrt{X{\left(\alpha \right)}^{2}+{\left(r_{1x}+r_{2x}\right)}^{2}}$$
where L(α) represents the cable length during elbow flexion-extension motion, and X(α) denotes the length of the current cable configuration onto the YOZ plane. $r_{1x}$ and $r_{2x}$ represent the x-axis component of vector $r_{1}$ and $r_{2}$, respectively. The parameter α corresponds to the elbow flexion angle, with a functional range of 0–120°. Through partial differentiation of the cable length equation with respect to α, the relationship between the rate of cable length variation (ΔL) and the elbow joint rotation angle (α) during flexion-extension motion can be established, expressed as:(8)$$\Delta L=\frac{\partial L(\alpha )}{\partial \alpha }$$

The parameters $r_{1YOZ}$ and $r_{2YOZ}$ are influenced by human body dimensions and must be determined through comprehensive consideration of both the human arm circumference and exoskeleton dimensions. Subsequently, based on Equation (Equation 8), the influence of $b_{1}$ and $b_{2}$ on the rate of cable length variation was investigated, with the corresponding relationship illustrated in Figure 5a. The analysis reveals that larger values of $b_{1}$ and $b_{2}$ result in greater cable length variation. However, when maintaining a constant angle between the driving cable and the Y-axis, increased values of $b_{1}$ and $b_{2}$ necessitate higher vertical positioning of the redirection points (B1 and B2) on the UAC, leading to excessive structural protrusion. Conversely, when maintaining a constant height of the redirection points (B1 and B2), larger values of $b_{1}$ and $b_{2}$ result in a decreased angle between the driving cable and the Y-axis, consequently reducing the effective moment arm. Through comprehensive optimization considering these factors, appropriate values for $b_{1}$ and $b_{2}$ were determined, yielding the cable kinematic results presented in Figure 5b.

### 3.2. Cable Dynamics

During elbow flexion-extension movements, cables S1 and S2 operate in coordination to provide actuating force. Based on the cable configuration and actuation mechanism of the exoskeleton robot, the cable dynamics in this study establish a mathematical relationship between cable tension and joint torque. The cable tension is denoted as $F_{i}\left(i=1,2\right)$. Based on the torque calculation formula for forces acting on a point in the spatial coordinate system, the resultant torque acting on point E (the elbow joint center) generated by the two cables can be calculated as follows:(9)$$M_{s}=\sum_{i=1}^{2}p_{i}\times F_{i}$$

Subsequently, projecting the torque $M_{s}$ onto the axes of elbow flexion-extension motion, the following relationship can be derived:(10)$$M_{s}=\sum_{i=1}^{2}p_{iy}F_{iz}$$
where $p_{iy}$ represents the y-axis component of vector $p_{i}$, and $F_{iz}$ denotes the z-axis component of force vector $F_{i}$. The required torque during elbow flexion-extension motion varies among individuals due to differences in joint angles and forearm mass, yet generally does not exceed 10 N·m [27]. Using the maximum torque value of 10 N·m as the design criterion, the resultant driving cable tension was calculated to be 512 N.

## 4. Motion Control Strategy

Patients in the early stages of paralysis, particularly those experiencing motor dysfunction due to spinal cord injuries, brain injuries, or severe neurological disorders, experience varying degrees of functional impairment. At this stage, patients lack autonomous movement capabilities, potentially leading to muscle atrophy, joint stiffness, and poor blood circulation. Therefore, it is essential to employ robotic exoskeletons to guide the affected limbs through predefined rehabilitation exercises, facilitating the reconnection between the limbs and the damaged central nervous system. This approach establishes a foundation for subsequent rehabilitation and neural recovery while mitigating pathological changes. Based on the cable-driven mechanism discussed in Section 3, a motion control strategy was designed for the passive motion of the elbow joint exoskeleton, as illustrated in Figure 6.

In the passive guidance mode, the flexion-extension trajectory of the elbow joint is predefined. Given that the system is intended for human rehabilitation and targets users with motor dysfunction, the exoskeleton must exhibit high safety and operational stability. However, rehabilitation exoskeletons must adapt to different patients, imposing stringent requirements on the stability and versatility of the control system. Variations in upper limb weight among patients result in different applied loads on the exoskeleton, which dynamically change with joint angles and postures during motion. Additionally, cables undergo elastic deformation under tension, leading to a nonlinear relationship between cable displacement and joint angles. These biomechanical characteristics, time-varying loads, contact uncertainties, and the effects of cable elasticity and friction can impact control precision.

To address these challenges, a speed control strategy was designed for the proposed cable-driven exoskeleton system. Here, $\alpha_{d}\left(t\right)$ and $l_{d}\left(t\right)$ represent the desired elbow joint angle and cable length, respectively, while $\alpha_{m}\left(t\right)$ and $l_{m}\left(t\right)$ denote the measured elbow joint angle and cable length. Notably, a PID controller is implemented in the proposed control scheme. Typically, the mathematical model of a PID controller is described as:(11)$$u\left(t\right)=K_{p}e\left(t\right)+K_{i}\int_{0}^{t}e\left(t\right)\;dt+K_{i}\frac{de(t)}{dt}$$
where u(t) represents the control output computed by the PID controller based on the error signal e(t), where $K_{p}$, $K_{i}$, and $K_{d}$ denote the proportional, integral, and derivative gain coefficients, respectively, for regulating the dynamic behavior of the system. For the designed system, the input variable corresponds to difference in cable length, while the control objective is the rotational speed of the brushless motor. Given that the motor directly controls the cable length, which indirectly governs the elbow joint angle, the relationship between cable length and joint angle exhibits a non-monotonic nonlinear characteristic. Therefore, the cable kinematics is employed to transform the elbow joint angle into the corresponding cable length, while the observed error is defined as the difference between the desired and actual cable length. This approach enables more precise control of the driving cables while effectively addressing the non-monotonic relationship between joint angle and cable length.

## 5. Experiments and Results

The prototype of the elbow exoskeleton system was established. To evaluate the effectiveness of the proposed elbow exoskeleton and its corresponding control strategy, physical experiments were conducted on six healthy subjects numbered S1 to S6 (all male, weight: 65 ± 5 kg, height: 1.73 ± 0.3 m, age: 23 ± 2 years). The experimental protocol has been granted by the Ethics Committee of Zhongda Hospital Southeast University, and each subject was given informed consent to participate in the experiments by signing a written agreement.

The experimental condition and process for the exoskeleton worn by a subject are shown in Figure 7. The subjects were required to sit on a chair, wear the exoskeleton, and fit the UAC and FAC of the exoskeleton onto their upper arm and forearm, respectively, with the arms naturally hanging down before the experiment began. The exoskeleton then moved according to the desired trajectory, completing five continuous motion cycles. Additionally, each subject performed five cycles of active movement at the same frequency without wearing the exoskeleton as a control group. In this system, the angular movement of the elbow joint is measured by two IMUs, placed on the UAC and FAC, respectively. The cable tensions are monitored by force sensors, and the sEMG signals from the biceps brachii and brachioradialis muscles were detected by sEMG sensors. To quantitatively analyze the error between the measured elbow joint angle and the desired elbow joint angle, root mean square error (RMSE), mean absolute error (MAE), and correlation coefficient (CC) are taken into account, which are defined as follows:(12)$$RMSE=\sqrt[{\;}]{\frac{1}{n}\sum_{i=1}^{n}{\left(\alpha_{i}-\hat{\alpha_{i}}\right)}^{2}}$$(13)$$MAE=\frac{1}{n}\sum_{i=1}^{n}\left|\alpha_{i}-\hat{\alpha_{i}}\right|$$(14)$$CC=\frac{\sum_{i=1}^{n}\left(\alpha_{i}-\bar{\alpha }\right)\left({\hat{\alpha }}_{i}-\bar{\hat{\alpha }}\right)}{\sqrt{\sum_{i=1}^{n}{\left(\alpha_{i}-\bar{\alpha }\right)}^{2}\sum_{i=1}^{n}{\left({\hat{\alpha }}_{i}-\bar{\hat{\alpha }}\right)}^{2}}}$$
where $\alpha_{i}$ and $\hat{\alpha_{i}}$ denote the measured and desired elbow motion angle, and n is the number of data points.

The PID parameters are predefined before the exoskeleton operation, with $K_{p}=49$, $K_{i}=1$, and $K_{d}=0.1$. Figure 8a illustrates the angular variation of the exoskeleton joint during the experiment conducted with subject S1. In this work, the desired trajectory is predefined, while the measured trajectory is obtained in real time from the IMU output. The error curve describes the deviation between the desired and actual angles during the movement. The desired and measured angles of the elbow joint follow a similar trend. Based on the experimental results from all subjects, the trajectory tracking errors between the desired and measured joint angles are calculated and summarized, as shown in Figure 9a. To evaluate the accuracy of trajectory tracking, the CC between the desired and measured motion angles is calculated, as shown in Figure 9b. The numerical results are shown in Table 1.

Due to the natural relaxation of the upper limb, the elbow joint does not extend fully. During the flexion-extension movement of the elbow joint, the range of motion varies from 10° to 120°. For the elbow joint, the trajectory of the exoskeleton-assisted movement closely matches the desired trajectory, with the CC remaining around 0.997, indicating very high tracking accuracy. However, it can be observed in Figure 8b that there is an overall delay in the trajectory tracking, with an average time of approximately 0.07 s. In this study, a Bowden cable is used for the distal drive to achieve energy and motion transmission. However, the Bowden cable has an inherent flaw, which is its susceptibility to delay. The Bowden cable typically consists of a steel cable and a plastic sheath. Due to the friction, dead zone, and hysteresis between the two components, delays occur as the steel wire moves within the sheath [28]. In addition, the inherent elasticity of the steel wire may cause deformation, leading to instability in the transmitted force. Furthermore, when the Bowden cable is bent or twisted, the driving force from the motor cannot be directly converted into linear motion, and the curvature of the steel wire induces a delay effect. Although trajectory delay is the primary cause of the motion angle tracking error, the overall tracking accuracy remains high. In rehabilitation training scenarios, the focus is mainly on the range of motion of the joint. According to the experimental results, the peak differences between the measured and desired trajectories are small, meeting the rehabilitation movement requirements. This indicates that the PID-based speed control technology applied in this exoskeleton system performs satisfactorily.

Observing the error curve in Figure 8c, it can be concluded that the error between the desired and measured motion angles of the elbow joint is confined within the range of −5° to 5°. Additionally, the variation in error is strongly correlated with the movement cycle. Since the elbow joint does not fully extend to 0° when the upper limb is in a relaxed hanging position, but rather maintains a certain angle, and the airbag attached to the forearm drives the forearm to bend by a certain angle after inflation, a delay phenomenon occurs when the elbow joint reaches its lowest point, resulting in larger errors at that time.

To qualitatively evaluate the assistive efficacy of an exoskeleton during elbow flexion and extension motions, sEMG sensors were employed to capture sEMG signals from the biceps brachii and brachioradialis muscles under two distinct conditions: unassisted voluntary movement and exoskeleton-assisted movement. The acquired sEMG signals were subjected to a fourth-order Butterworth bandpass filter with a frequency range of 20–450 Hz to attenuate low-frequency artifacts, such as baseline drift and high-frequency noise, including electrical interference. Following this, the signals underwent full-wave rectification and were smoothed using a zero-lag 10 ms moving average filter to derive the envelope signals. These envelope signals were then normalized based on the sEMG amplitude, enabling the computation of integrated electromyography (iEMG) across different movement cycles for each subject.

Figure 10 depicts the sEMG envelope signals of the biceps brachii and brachioradialis muscles, alongside the tension forces of the two cables, for subject S1 during both exoskeleton-assisted passive movement and unassisted voluntary movement. The data reveal that during voluntary movement, the cables remained in a relaxed state, devoid of tension, and the sEMG signal amplitudes were notably higher. Conversely, during exoskeleton-assisted passive movement, the cables exhibited measurable tension, and the sEMG signal amplitudes were substantially reduced. These findings suggest that the exoskeleton effectively facilitates passive elbow joint motion and compensates for a significant portion of the muscular exertion required. The experimental outcomes for six participants are illustrated in Figure 11.

A marked reduction in both iEMG values for the biceps brachii and brachioradialis muscles was observed when the exoskeleton was worn, with decreases of 37.7% and 28.8%, respectively. This empirical evidence substantiates the utility of the exoskeleton in enhancing motor assistance during therapeutic interventions.

## 6. Conclusions

This paper proposed a shrimp-inspired robotic exoskeleton for elbow rehabilitation of patients with upper-limb dysfunction. The biomimetic design and mechatronic system of this exoskeleton were systematically introduced, in which Bowden cables were employed for remote actuation of the exoskeleton. Comprehensive modeling and analysis of cable kinematics and dynamics were conducted based on the mechanical design and cable configuration of the exoskeleton, and a PID-based position control strategy was then developed accordingly. To investigate the exoskeleton-assisted performance in passive movement of the elbow joint, experimental studies involving six subjects were carried out based on a prototype of the exoskeleton. The experimental results indicate that the measured trajectory of the elbow joint closely follows the desired trajectory, ensuring smooth rehabilitation movements. Compared to the condition without an exoskeleton, obvious reductions in iEMG values of the biceps brachii and brachioradialis muscles can be observed under the condition with an exoskeleton, demonstrating the effectiveness of the exoskeleton system in elbow assistance.

To further enhance the accuracy of the control strategy, future work will focus on reducing temporal delays in exoskeleton-assisted trajectories through error compensation techniques. Meanwhile, to validate the applicability of the exoskeleton for patients with muscle stiffness and to expand its training modes, future research will focus on conducting experimental evaluations involving target patient populations while incorporating active assistance and resistance training modes. In addition, cushioning airbags will be integrated into the lower part of the exoskeleton to prevent potential emergencies such as sudden cable rupture.

## Figures and Tables

**Figure 1 biomimetics-10-00271-f001:**
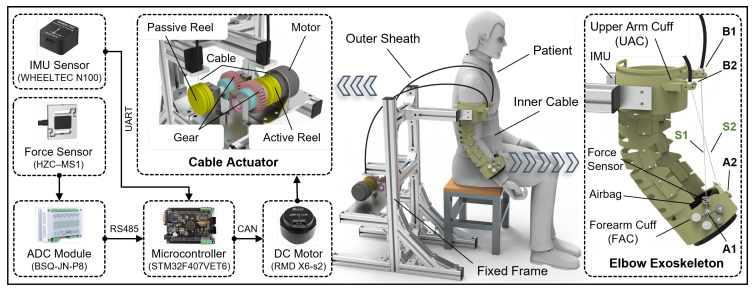
Mechatronic system of the elbow exoskeleton and schematic illustration of human-exoskeleton coupling rehabilitation. Cables S1 and S2 pass through anchor points B1 and B2 on the UAC and are then secured to anchor points A1 and A2 on the FAC.

**Figure 2 biomimetics-10-00271-f002:**
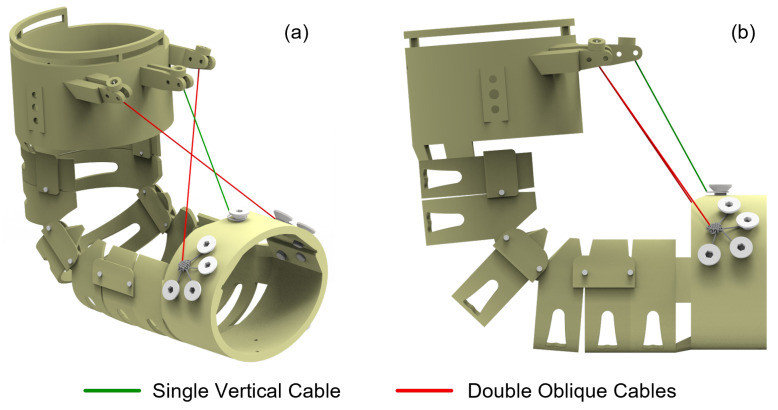
Comparison of single vertical and double oblique cable layouts for the elbow exoskeleton. (**a**) Isometric view. (**b**) Side view.

**Figure 3 biomimetics-10-00271-f003:**
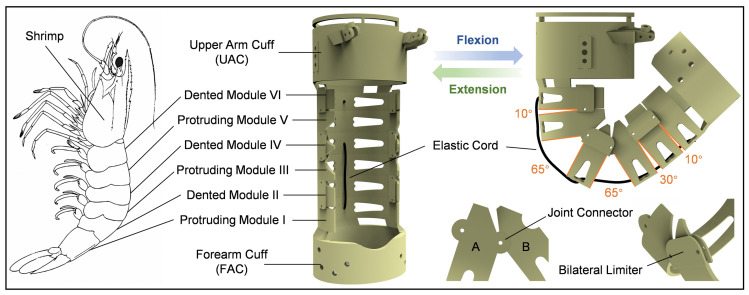
Biomimetic mechanical design of the elbow exoskeleton inspired by the structural characteristics of shrimp shell.

**Figure 4 biomimetics-10-00271-f004:**
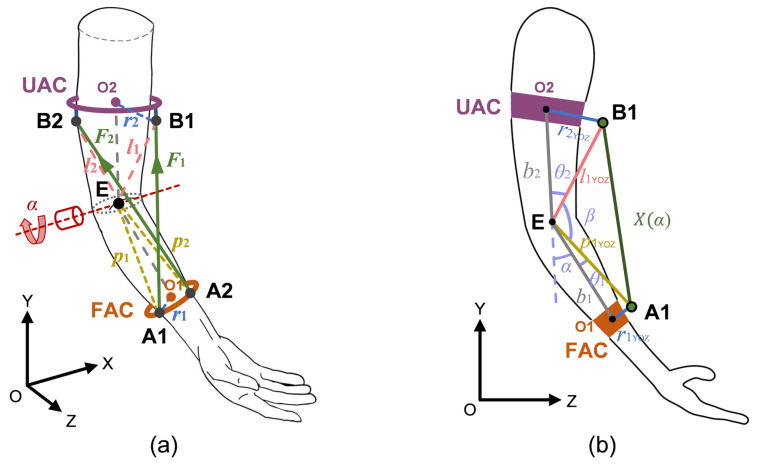
Schematic illustration of the elbow exoskeleton. (**a**) Simplified model of cable arrangement. (**b**) Kinematic model of human-exoskeleton coupling.

**Figure 5 biomimetics-10-00271-f005:**
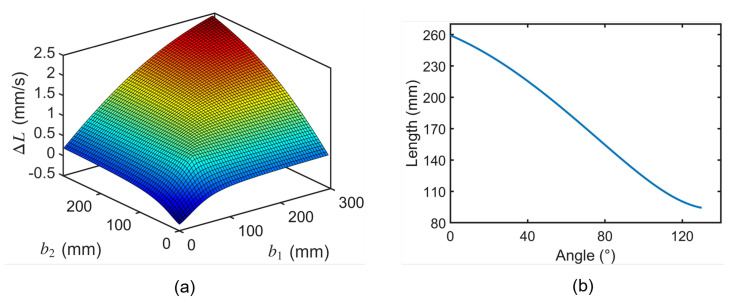
Analysis of cable kinematics. (**a**) Change rate of cable length. (**b**) Relationship between cable length and elbow joint angle.

**Figure 6 biomimetics-10-00271-f006:**
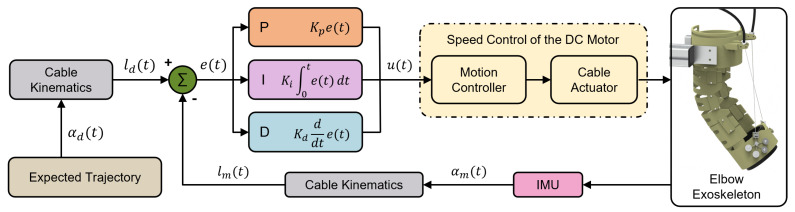
Block diagram of the PID-based control strategy for elbow motion assistance with the proposed exoskeleton.

**Figure 7 biomimetics-10-00271-f007:**
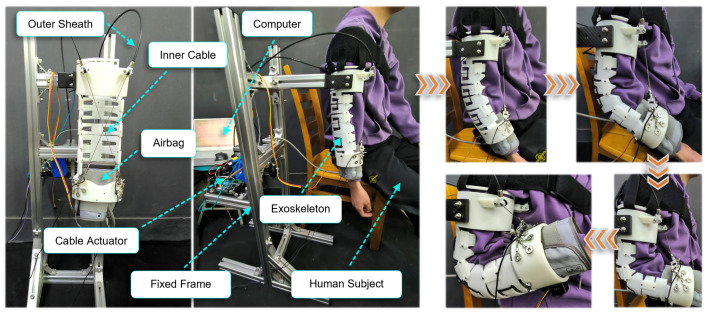
Experimental condition of the elbow exoskeleton system and experimental process of the exoskeleton-assisted motion for a human subject.

**Figure 8 biomimetics-10-00271-f008:**
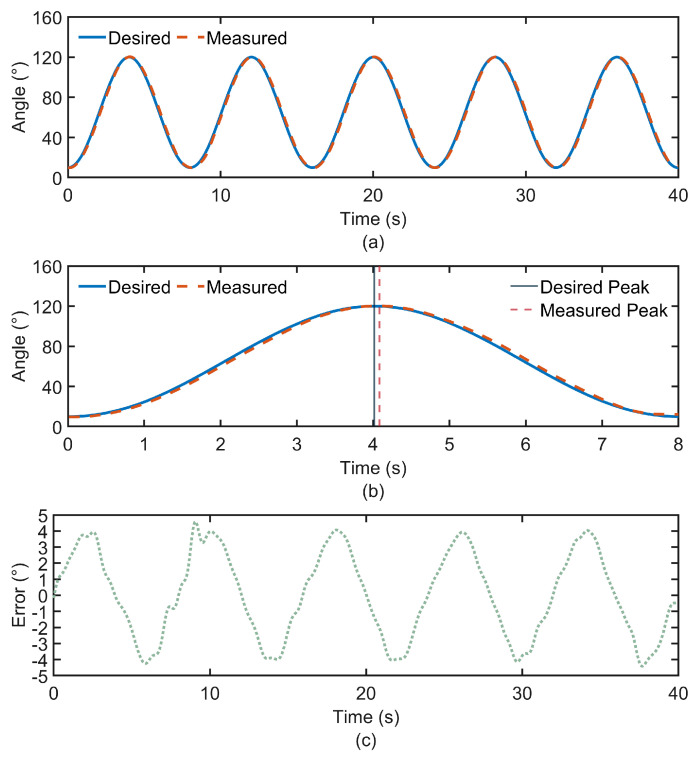
Variations in the motion angles of the elbow joint. (**a**) Desired and measured trajectories. (**b**) Time delay of the measured trajectory relative to the desired trajectory. (**c**) Position tracking error between desired and measured trajectories.

**Figure 9 biomimetics-10-00271-f009:**
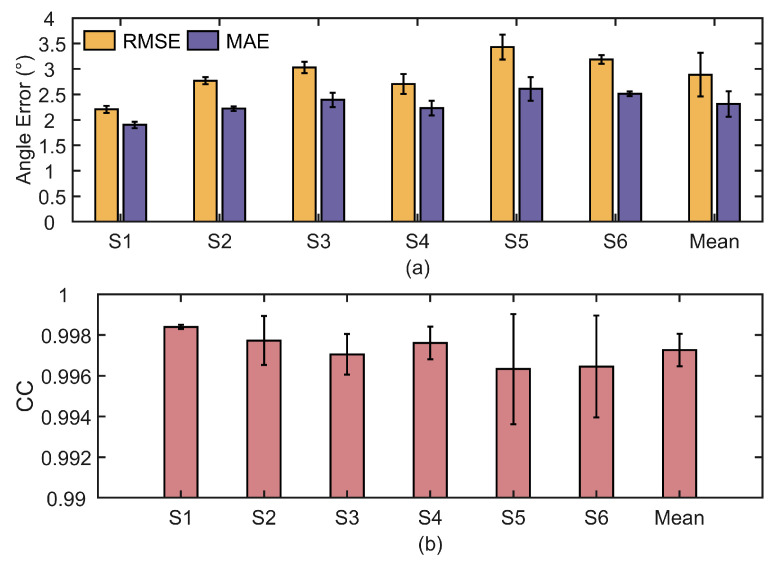
Statistical results of position control experiments (mean ± SD). (**a**) Position tracking errors (RMSE and MAE). (**b**) Position tracking accuracy (CC).

**Figure 10 biomimetics-10-00271-f010:**
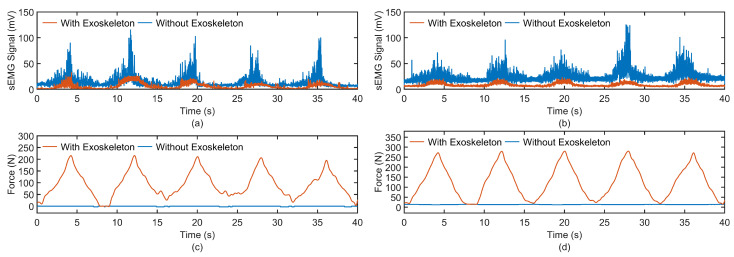
Variations in sEMG signals and cable forces under conditions with and without the elbow exoskeleton. sEMG signals of (**a**) biceps brachii and (**b**) brachioradialis. Tension force of (**c**) cable 1 (**d**) cable 2.

**Figure 11 biomimetics-10-00271-f011:**
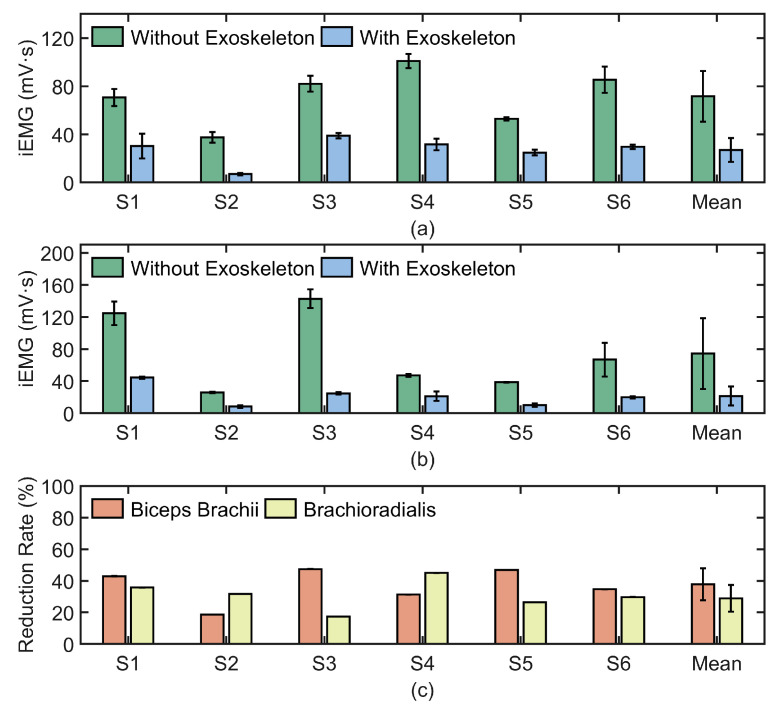
iEMG results of all subjects (mean ± SD). (**a**) Biceps brachii. (**b**) Brachioradialis. (**c**) Reduction rate of iEMG.

**Table 1 biomimetics-10-00271-t001:** Statistical results of position control experiments in different subjects.

Subject	RMSE (°)	MAE (°)	CC
S1	2.207 ± 0.069	1.901 ± 0.065	0.9984 ± 0.0001
S2	2.767 ± 0.069	2.220 ± 0.044	0.9977 ± 0.0012
S3	3.028 ± 0.111	2.392 ± 0.142	0.9971 ± 0.0010
S4	2.704 ± 0.194	2.232 ± 0.142	0.9976 ± 0.0008
S5	3.429 ± 0.242	2.607 ± 0.232	0.9963 ± 0.0027
S6	3.188 ± 0.086	2.512 ± 0.045	0.9965 ± 0.0025
Mean	2.887 ± 0.428	2.311 ± 0.252	0.9973 ± 0.0008

## Data Availability

The data used to support the findings of this study are available from the corresponding author upon reasonable request.

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
