# Peer review of "Biomimetic Design and Validation of an Adaptive Cable-Driven Elbow Exoskeleton Inspired by the Shrimp Shell"

_biomimetics, 2025, doi:10.3390/biomimetics10050271_

Round 1
Reviewer 1 Report
Comments and Suggestions for Authors
- The content of this paper is complete and the logic is clear. However, the introduction part can further compress the redundant background introduction and focus on the unique advantages and research gaps of bionic cable-driven technology.
-
In Section 5, line 321, it is mentioned that "Observing the error curve, it can be concluded that the error between the desired and measured motion angles of the elbow joint is confined within the range of -5° to 5°."
In line 321, it is also mentioned that "However, it can be observed that there is an overall delay in the trajectory tracking, with a delay time of approximately 0.07 s."
However, I cannot see this from Figures 7 and 8. It is recommended to modify the scale, annotations, and clarity of the graphs to display the experimental results more intuitively. -
The current experiments were only conducted on healthy subjects, and no plan for future validation in patient populations was specified. It is recommended to discuss the applicability of the exoskeleton for patients with muscle stiffness in the future.
-
The experiment only tested passive motion and did not consider situations such as active assistance, resistance training, and sudden cable rupture. It is recommended to discuss more application scenarios and emergency mechanisms in the future to enhance the safety evaluation.
Author Response
We sincerely appreciate your valuable feedback and insightful suggestions. Your thoughtful review has greatly contributed to improving the quality of our manuscript. We have made revisions based on your comments, with red strikethrough text indicating deleted content and blue text highlighting additions in the revised version. The responses to all of your comments are listed as follows:
Comment 1: The content of this paper is complete and the logic is clear. However, the introduction part can further compress the redundant background introduction and focus on the unique advantages and research gaps of bionic cable-driven technology.
Response to Comment 1: Based on your helpful suggestion, we have compressed the redundant background introduction on upper limb dysfunction. It is now more concise and mainly focused on the key findings and contributions of our research work. The introduction has been restructured to concentrate on cable-driven exoskeletons, emphasizing both their strengths and the challenges they present, which in turn provides the motivation for our study.
Comment 2: In Section 5, line 321, it is mentioned that "Observing the error curve, it can be concluded that the error between the desired and measured motion angles of the elbow joint is confined within the range of -5° to 5°."
In line 321, it is also mentioned that "However, it can be observed that there is an overall delay in the trajectory tracking, with a delay time of approximately 0.07 s."
However, I cannot see this from Figures 7 and 8. It is recommended to modify the scale, annotations, and clarity of the graphs to display the experimental results more intuitively.
Response to Comment 2: Based on your helpful suggestion, we have redrawn Figure 7 and added Figures 7b and 7c to better illustrate the time delay of the measured trajectory relative to the desired trajectory, as well as the position tracking error between the desired and measured trajectories.
Comment 3: The current experiments were only conducted on healthy subjects, and no plan for future validation in patient populations was specified. It is recommended to discuss the applicability of the exoskeleton for patients with muscle stiffness in the future.
Response to Comment 3: Based on your helpful suggestion, we have added to the Conclusion section that future work will involve experimental evaluations with patients, in order to assess the applicability of the exoskeleton for individuals with upper limb dysfunction.
Comment 4: The experiment only tested passive motion and did not consider situations such as active assistance, resistance training, and sudden cable rupture. It is recommended to discuss more application scenarios and emergency mechanisms in the future to enhance the safety evaluation.
Response to Comment 4: Based on your helpful suggestion, we have added to the Conclusion section that future work will focus on developing additional motion patterns and designing an emergency mechanism to address potential sudden cable rupture.
Lastly, we hope the revisions meet your expectations and effectively address your concerns. Thank you again for your time and effort.
Reviewer 2 Report
Comments and Suggestions for Authors
The article presents the design of an elbow exoskeleton inspired by the shrimp shell. I found their approach novel, as I have not encountered any other research that addresses the problem of joint misalignment in this way. Thus, I recommend the article for publication after the following minor changes:
-
Please add a figure describing the configuration of the Bowden cables, as it is difficult to visualize the relationship between the two cables and how they contribute to the system's improvement.
-
The authors mention that they implemented an optimization method to determine the anchor points of the Bowden cables. However, neither the objective function nor the optimization method is described. Please include this information.
-
When explaining the control strategy, you mention that the error is calculated by comparing the desired angle with the measured angle to bypass nonlinearities. However, the control diagram presented in Fig. 5 shows the error as the difference between cable lengths. Please clarify or correct this inconsistency.
Author Response
We sincerely appreciate your valuable feedback and insightful suggestions. Your thoughtful review has greatly contributed to improving the quality of our manuscript. We have made revisions based on your comments, with red strikethrough text indicating deleted content and blue text highlighting additions in the revised version of our manuscript. The responses to all of your comments are listed as follows:
Comment 1: Please add a figure describing the configuration of the Bowden cables, as it is difficult to visualize the relationship between the two cables and how they contribute to the system's improvement.
Response to Comment 1: Based on your helpful suggestion, we have added Figure 2 to more clearly illustrate the configuration of the double oblique cables. By comparing it with the single vertical cable, It shows that the double oblique cables are positioned closer to the limb surface. Additionally, we have included an explanation in the text that the symmetrical configuration of the double oblique cables allows the lateral tensile forces to mutually cancel out, thereby enhancing the overall stability of the system.
Comment 2: The authors mention that they implemented an optimization method to determine the anchor points of the Bowden cables. However, neither the objective function nor the optimization method is described. Please include this information.
Response to Comment 2: We gratefully appreciate for your valuable suggestion, and we understand your concerns. During the design, we considered the kinematic analysis and the influence of the lengths of b₁ and b₂ on the structure, and manually selected anchor point positions. We believe the ambiguity arose from the previous wording that referred to the “optimal” anchor point positions, which has now been removed for clarity.
Comment 3: When explaining the control strategy, you mention that the error is calculated by comparing the desired angle with the measured angle to bypass nonlinearities. However, the control diagram presented in Fig. 5 shows the error as the difference between cable lengths. Please clarify or correct this inconsistency.
Response to Comment 3: Based on your helpful suggestion, we have corrected the previous misstatement regarding the term error in the manuscript. It has been revised to clarify that error refers to the difference between the cable lengths.
Lastly, we hope the revisions meet your expectations and effectively address your concerns. Thank you again for your time and effort.
Reviewer 3 Report
Comments and Suggestions for Authors
The research proposes an elbow exoskeleton inspired by a shrimp shell which solves joint misalignment in a fixed-frame clinical setting. The kinematics, dynamics, and control scheme presented are sound, however, I suggest few changes to the scope and revisions for clarity.
In the current state, the emphasis of the paper should be shifted away from biomimicry and more toward bio-inspiration, as the discussion in section 2.2 is sparse. I would like to see angle comparisons to the biological dented/protrusion modules to indicate biomimicry.
Some changes for clarity are as follows. Labels should be specified in the caption of figure 1. Color choices appear very pale in figure 3 and should be changed to be darker. Figure 6 should have bolded arrows that contrast with the background for the labels. The entire paper should be proofread for spelling errors.
Author Response
We sincerely appreciate your valuable feedback and insightful suggestions. Your thoughtful review has greatly contributed to improving the quality of our manuscript. We have made revisions based on your comments, with red strikethrough text indicating deleted content and blue text highlighting additions in the revised version. The responses to all of your comments are listed as follows:
Comment 1: In the current state, the emphasis of the paper should be shifted away from biomimicry and more toward bio-inspiration, as the discussion in section 2.2 is sparse. I would like to see angle comparisons to the biological dented/protrusion modules to indicate biomimicry.
Response to Comment 1: We gratefully appreciate for your valuable suggestion, and we understand your concerns. Our research focuses on the design of an elbow exoskeleton inspired by the structural characteristics of the shrimp shell. Based on your helpful suggestion, we have added a structural comparison between our exoskeleton and the shrimp shell in Section 2.2, highlighting that our design is inspired by the morphology of the shrimp. Similar to the segmented abdominal structure of a shrimp, our exoskeleton consists of six modules. The joints formed between adjacent modules allow a certain degree of angular movement, enabling the system to achieve a high level of bending flexibility, analogous to that of the abdomen of shrimp.
Comment 2: Some changes for clarity are as follows. Labels should be specified in the caption of figure 1. Color choices appear very pale in figure 3 and should be changed to be darker. Figure 6 should have bolded arrows that contrast with the background for the labels. The entire paper should be proofread for spelling errors.
Response to Comment 2: Based on your helpful suggestion, we have updated the caption of Figure 1 to include descriptions of the labels for better clarity. The colors in Figure 3 have been enhanced to improve visual clarity. Additionally, the arrows in Figure 6 have been changed to a color that contrasts more clearly with the background to enhance their visibility.
Lastly, we hope the revisions meet your expectations and effectively address your concerns. Thank you again for your time and effort.